# Generation of Tailored Extracellular Matrix Hydrogels for the Study of In Vitro Folliculogenesis in Response to Matrisome-Dependent Biochemical Cues

**DOI:** 10.3390/bioengineering11060543

**Published:** 2024-05-25

**Authors:** Hannah B. McDowell, Kathryn L. McElhinney, Elizabeth L. Tsui, Monica M. Laronda

**Affiliations:** 1Stanley Manne Children’s Research Institute, Ann & Robert H. Lurie Children’s Hospital of Chicago, Chicago, IL 60611, USA; hannah.mcdowell@northwestern.edu (H.B.M.);; 2Department of Pediatrics, Division of Endocrinology, Feinberg School of Medicine, Northwestern University, Chicago, IL 60611, USA; 3Department of Surgery, Feinberg School of Medicine, Northwestern University, Chicago, IL 60611, USA; 4Department of Obstetrics and Gynecology, Feinberg School of Medicine, Northwestern University, Chicago, IL 60611, USA

**Keywords:** extracellular matrix, hydrogel, bioprosthetic ovary, follicle culture

## Abstract

While ovarian tissue cryopreservation (OTC) is an important fertility preservation option, it has its limitations. Improving OTC and ovarian tissue transplantation (OTT) must include extending the function of reimplanted tissue by reducing the extensive activation of primordial follicles (PMFs) and eliminating the risk of reimplanting malignant cells. To develop a more effective OTT, we must understand the effects of the ovarian microenvironment on folliculogenesis. Here, we describe a method for producing decellularized extracellular matrix (dECM) hydrogels that reflect the protein composition of the ovary. These ovarian dECM hydrogels were engineered to assess the effects of ECM on in vitro follicle growth, and we developed a novel method for selectively removing proteins of interest from dECM hydrogels. Finally, we validated the depletion of these proteins and successfully cultured murine follicles encapsulated in the compartment-specific ovarian dECM hydrogels and these same hydrogels depleted of EMILIN1. These are the first, optically clear, tailored tissue-specific hydrogels that support follicle survival and growth comparable to the “gold standard” alginate hydrogels. Furthermore, depleted hydrogels can serve as a novel tool for many tissue types to evaluate the impact of specific ECM proteins on cellular and molecular behavior.

## 1. Introduction

Advancements in anticancer therapies have improved survival rates of pediatric and young adult cancer patients, leading to a growing population of long-term survivors [1,2,3]. Despite improved prognoses, life-saving treatments can result in loss of fertility and gonadal endocrine function, resulting in an increased risk of developing comorbidities such as osteoporosis, cardiovascular disease, and declining cognitive function [4,5,6,7]. Consequently, fertility preservation has increasingly become a priority for patients and physicians [8,9]. However, conventional fertility preservation options are not feasible for pre-pubertal children as they cannot undergo ovarian stimulation and egg retrieval. Within the last decade, the number of individuals choosing to undergo ovarian tissue cryopreservation (OTC) and ovarian tissue transplantation (OTT) has increased [10]. OTT of OTC tissue seeks to fully recover endocrine and reproductive function, thereby reducing long-term comorbidities and expanding patients’ reproductive options.

Transplanted tissue functionality is directly correlated with the number of primordial follicles (PMFs) that comprise the ovarian reserve. There is rapid activation and subsequent depletion of PMFs shortly after transplantation of the ovarian tissue, which results in reduced tissue functionality [11,12]. Additionally, patients with known malignancy or metastases in the ovary are not eligible for transplantation of ovarian tissue in its native form [13]. Endocrine function is restored in 95% of patients who undergo OTT. However, the average duration of endocrine function is approximately 2–5 years [14,15,16]. While hormone replacement therapies are viable alternatives for some individuals with ovaries, options that more closely mimic physiologic endocrine function are desired [17]. In addition, over 140 live births post-OTT have been documented, but the live birth rate remains between 20 and 40% [10,18]. Thus, opportunities to improve OTC and OTT for more individuals are drivers for ongoing research. Consequently, to extend transplanted tissue longevity and expand options for ineligible patients, the development of a bioengineered ovary that supports long-term follicle growth and endocrine function is critical.

In recent years, research has been conducted to develop a three-dimensional (3D) scaffold that would mimic the microenvironment of the ovary. Previous work has shown that the use of a 3D-printed gelatin scaffold seeded with isolated follicles restores hormone production and fertility in ovariectomized mice [19]. However, current methods such as these pose several challenges when moving from the bench to the clinic. Specifically, under typical circumstances, PMFs remain quiescent, and follicular development occurs in waves throughout the reproductive timeline, allowing for long-term fertility and endocrine function [20]. Yet during OTT, PMFs are rapidly depleted. Although the mechanism by which PMFs are activated post-OTT is unknown, it has been suggested that post-transplantation reperfusion ischemia plays a role [12]. To address this issue, there have been numerous attempts to increase vascularization into tissue and diffusion into scaffolds by utilizing porous materials, designing specific pore geometries with 3D printed scaffolds, and including angiogenic factors within the biomaterial [19,21,22,23]. However, developing a fully functionalized scaffold that meets the precise parameters of the human ovary will be necessary for clinical applications. Notably, the human ovary is compartmentalized into two regions: the cortex and the medulla, with considerable gross histological differences between the regions [24,25]. PMFs are primarily located within the cortex of the ovary and require specific physical and biochemical parameters to remain quiescent. As such, it will be important to consider the physical, chemical, and mechanical properties of each compartment to engineer a bioprosthetic ovary with long-term function.

Hydrogels built from decellularized organs could support or be tuned to match organ physical, chemical, and mechanical properties [26]. These hydrogels retain native extracellular matrix (ECM) composition of the desired organ, are biocompatible, and preserve biological signaling [27]. In this study, bovine ovaries were used as they are mono-ovulatory animals and exhibit similar compartmentalization when compared to human ovaries. The utilization of bovine dECM is highly translatable for clinical purposes as bovine products have been FDA-approved and are often used for regenerative medicine and transplantations. Within the bovine ovary, the cortex is ≈8.5 times more rigid than the medulla and is comprised of different matrisome proteins [28]. The matrisome, the contingent of genes encoding proteins that comprise the ECM and ECM-related proteins, provides integral structural and physical support as well as necessary signaling cues that control tissue-specific cellular behavior [29,30,31]. It is hypothesized that differential matrisome composition, density, and physical attributes in the specified ovarian regions lead to different cellular behaviors that are critical for developing bioengineering scaffolds that influence folliculogenesis.

Follicles grown in vitro prefer a three-dimensional (3D) encapsulated environment and are well-supported in a bioinert hydrogel, alginate [32]. To study the effects of matrisome proteins that exist within the two main compartments of the ovary on folliculogenesis, we sought to develop hydrogels that are derived from the cortex or medulla dECM and selectively deplete POIs from each compartment-specific hydrogel. To validate our method of selective POI depletion in dECM hydrogels, we chose two POIs shown to be differentially abundant in the cortex and medulla, elastin microfibril interface-located protein 1 (EMILIN1) and zona pellucida protein 3 (ZP3). EMILIN1 is a secreted glycoprotein that is localized within the immediate microenvironment of PMF in porcine and human ovaries [33,34]. Additionally, we chose to deplete a matrisome protein that is only found in the ovary, ZP3, which surrounds oocytes and is in contact with adjoining granulosa cells. The main goal of this methodology is to effectively deplete dECM hydrogels of a specific POI. We also sought to define the effect of depletion on the physical and architectural properties of the hydrogels.

We have developed methods for generating optically clear ovarian dECM hydrogels derived from bovine cortical tissue (CTX dECM) and medullary tissue (MED dECM), which can be selectively depleted of proteins of interest (POI). With these tools, folliculogenesis can be studied in tailored environments to understand a specific matrisome protein’s role on follicle quiescence, activation, growth, ovulation, and hormone production. Additionally, this work lays the groundwork for designing an enhanced bioengineered scaffold for a bioprosthetic ovary.

## 2. Materials and Methods

### 2.1. Obtaining and Processing Bovine Ovaries 

Post-pubertal bovine ovaries were purchased from Applied Reproductive Technology, LLC (ART, Madison, Wisconsin, USA). Exact age ranges for the cattle at time of sacrifice and retrieval were not available. Ovaries were shipped at 4 °C overnight and were used within 24 h of retrieval. Only grossly normal ovaries were used for experiments. Bovine specimens were known to be from cycling cows due to the presence of large antral follicles and corpus lutea. To generate compartment-specific decellularized bovine ovarian tissue, ovaries were bisected through the hilum and sectioned sagittally into 0.5 mm slices starting from the ovarian surface epithelium. To generate consistent and precise slices, a 0.5 mm Thomas Stadie-Riggs Tissue Slicer (Thomas Scientific, Logan Township, NJ, USA) was used. The first 0.5 mm slice was designated as the cortex, the 1.0 mm slice was discarded to ensure clear cortex and medulla compartmentalization, and slices 1.5–3.0 mm were then designated as the medulla. 

### 2.2. Decellularization of Bovine Tissue

Decellularization was carried out using 0.1% sodium dodecyl sulfate (SDS) in 1× phosphate-buffered saline (PBS) [33,35]. For hydrogel fabrication, approximately 10 bovine ovaries were pooled for each biological replicate. Isolated cortical and medullary tissue was placed in a glass bottle with 500 mL of 0.1% SDS and a magnetic stir bar on a stir plate at 4 °C for 72–96 h. SDS solution was changed every 24 h for both conditions. Post-decellularization, the tissue was rinsed 3 times for 5 min in double distilled H_2_O. 

### 2.3. Ovarian Hydrogel Fabrication (Appendix A)

Decellularized CTX or MED ovarian tissue was frozen at −80 °C overnight and then lyophilized using a Freezone 6 Plus System. After lyophilization, tissue was milled through a 60 μm mesh screen using a mini tabletop mill (Eberbach E3300.00 single speed mini cutting mill 115 V). Three 24 h 100% ethanol (EtOH) washes were used to remove residual SDS and lipids that would interfere with gelation. Every 24 h, EtOH was exchanged. Washed, milled tissue was subsequently frozen at −80 °C and lyophilized using the Freezone 6 Plus System. To generate hydrogels, 5 mg/mL of tissue was digested with 1 mg/mL pepsin in 0.1 M hydrochloric acid (HCl) at a pH of ≈1–2, for 72 h at room temperature with agitation. After digestion, the solution was neutralized using 1 M sodium hydroxide (NaOH) and 10× PBS to ensure proper physiological pH and salinity. To neutralize, a 1:10 dilution is most often appropriate to bring the hydrogel back to suitable salinity and pH. For example, to produce 4 mL of dECM hydrogel, 400 μL of 10× PBS, and 400 μL of 1 M NaOH can be added to form the final neutralized hydrogel solution of 4.16 mg/mL. For follicle encapsulation, 150 μL of hydrogel is mixed with 50 μL of 2% alginate to generate a final dECM hydrogel spiked with alginate, with a final alginate concentration of 0.5%.

### 2.4. Magnetic Assisted Protein Filtration of Ovarian Hydrogels (Appendix A)

Post-neutralization but pre-gelation, dECM hydrogels can be depleted of a specific POI using magnetic-assisted protein filtration (MAPF). A total of 2 mg of sheep anti-rabbit IgG magnetic beads (Ref: 11203D; lot: 2686842, ThermoFisher Scientific) were aliquoted into 1.5 mL Eppendorf tubes and washed three times with 1 mL of cold 1× PBS. Washes were conducted using a magnetic separation rack to separate the beads, and then the supernatant was discarded. A total of 3 μg of polyclonal anti-rabbit EMILIN1 antibody (PA5-51745; lots: XK3739639B and YH4015124B, ThermoFisher Scientific) or ZP3 (A13156; lot: 1155930201, Abclonal) was added to the magnetic beads in 480 μL of PBS and allowed to nutate at 4 °C for 1 h. After incubation, POI antibody-bound magnetic beads were washed three times with 1 mL of cold 1× PBS and resuspended in 200 μL of 1× PBS. A total of 100 μL of POI antibody-bound magnetic beads were added to neutralized CTX and MED dECM hydrogels and allowed to nutate at room temperature for 1 h. Importantly, the hydrogel was neutralized but not allowed to gel (no 37 °C incubation was performed at this stage). After incubation with primary antibody-coated magnetic beads, the hydrogel suspension was passed through a magnetic column (Cat: 130-042-201; Lot: 9221110472 and lot: 9221200068, Miltenyi Biotec). The hydrogel that passed through the column was collected in 1.5 mL Eppendorf tubes and placed on a magnetic separation rack to pull out any remaining beads and the fully depleted supernatant is collected.

### 2.5. Nanoindentation for Determining dECM Hydrogel Rigidity

Nanoindentation (NI) to characterize the physical properties of materials was performed using the Piuma nanoindenter. Before NI testing, 250 μL of each hydrogel condition was plated on a 60 mm tissue-culture-treated Petri dish and incubated at 37 °C for 1 h to promote gelation. Analyses for all materials were carried out using a probe manufactured by Optics 11 with the following properties: Geofactor in air (3.89), stiffness (0.024 N/m), and tip radius (27.5 µm)—#P230090M. The probe was first calibrated in PBS against a 60 mm tissue-culture-treated Petri dish. PBS was then added to each Petri dish containing engineered materials. The Piuma software (V3.2.0) found the surface of the material and acquired indentation data. Data acquisition was carried out using a 0.03 threshold, 10 µm step-size, and a 5 s hold for adhesion mode.

### 2.6. Evaluation of Collagen Fiber Assembly and Architecture Using Second-Harmonic Generation Microscopy (SHG) 

Before imaging, 250 μL of each hydrogel condition was plated in a 60 mm tissue-culture-treated Petri dish and incubated at 37 °C for 1 h. Using a Nikon A1R-Multiphoton microscope, hydrogels and depleted hydrogels were imaged at 800 nm, 920 nm, and 950 nm to assess fiber organization. For SHG images displayed in this work (Appendix A), NIS Elements Viewer 5.21 64-bit software was used to artificially modify the pseudo color of the image from “blue” to “green fire blue” to highlight the intensity gradient of the fibers. Fiber orientation analysis was performed using CT-FIRE V3.0 Beta, a Matlab-based open-source software. Fiber width, length, straightness, and angle were determined for ten 1024-pixel regions per hydrogel condition for each experiment.

### 2.7. Jess Simple Western for Determination of Protein Depletion

A total of 100 μL of hydrogel from CTX and MED dECM before depletion was collected to determine baseline protein concentration. After depletion, 100 μL of each material was collected. Finally, the magnetic beads were collected to assess levels of bound POI. Protein abundance within the engineered hydrogels was determined using the Jess Simple Western System. Protein concentration was measured using a BCA assay according to manufacturer’s protocol (ThermoFisher Scientific). Following the Jess Western protocol, we quantified the relative concentration of COL4 (ab6586; lot: GR3448024-4, Abcam), EMILIN1 (ab231052; lot: GR3359624-1 and lot: GR3397745-1, Abcam), and ZP3 (A13156; lot: 1155930201, Abclonal) within all hydrogel conditions. Analysis of protein abundance was determined using Compass for SW software version 6.3.0. Using the electropherogram outputs, we calculated the area under the curve for the proteins of interest to assess the quantitative value for protein abundance. Representative Jess Western Blots for dECM hydrogels and depleted conditions can be found in Appendix A.

### 2.8. Primary Murine Follicle Encapsulation and Culture in POI-Depleted dECM Hydrogels

To assess the ability for engineered hydrogels to support follicle growth and survival, primary murine follicles (≈115 μm) were encapsulated in groups of 10 and cultured in dECM hydrogels and hydrogels depleted of EMILIN1 for 6 days. Follicle growth studies were conducted with 5 culture conditions: 0.5% Alginate (control), CTX dECM, MED dECM, CTX dECM depleted of EMILIN1, and MED dECM depleted of EMILIN1. All hydrogels were spiked with alginate to a final concentration of 0.5% as described above. Animal use was performed under a Northwestern University Animal Use and Care Committee-approved protocol. Prior to encapsulation, ovaries were removed from post-natal day 12–14 CD1 mice (Charles River Laboratories). Animals were housed in a temperature- and light-controlled environment (12 h light/12 h dark) and provided with food and water ad libitum. Follicles were isolated from the ovary using previously published mechanical isolation methods [36]. High-quality single-layered primary follicles were selected for long-term follicle culture. Ten primary follicles were encapsulated in control 0.5% alginate or spiked dECM hydrogels by crosslinking in CaCl_2_ solution (50 mM CaCl_2_, 140 mM NaCl, in sterile H_2_O). After crosslinking, the bead was moved to a 96-well plate containing 150 μL of follicle growth media (FGM) containing (α-MEM, 1 mg/mL fetuin, 1 mg/mL bovine serum albumin (BSA), 5 μm insulin, 5 μg/mL transferrin and 5 ng/mL selenium, and 10 mIU/mL recombinant human-follicle-stimulating hormone). Encapsulated follicles recovered for an hour in the incubator at 37 °C before imaging. Every other day, half of the spent media was replaced, and images were taken until the culture endpoint to observe health and determine follicle diameters. Follicle diameter was determined using ImageJ, and the diameter along an x- and y-axis was averaged for each follicle with a clear basement membrane. Of note, we observed a rupturing phenotype when follicles were grown in dECM hydrogels. Granulosa cells expanded out, creating a bulbous follicle where the oocyte was surrounded by cells but no longer had a clear and distinct basement membrane. For survival calculations, ruptured follicles were classified as “viable” until the oocyte was more than 50% exposed.

## 3. Results

### 3.1. dECM Hydrogels Derived from the Cortex or Medulla of Bovine Ovaries Had Similar Physical and Architectural Characteristics

The rigidities of the hydrogels from the cortical or medullary compartments were assessed using nanoindentation. Unlike bovine ovarian tissue, CTX and MED dECM hydrogels fabricated with 5 mg/mL of protein were similarly rigid (Appendix A). Using images from second harmonic generation (SHG) microscopy, we also assessed the fiber organization within the hydrogels and found no significant differences between the fiber angle, length, width, or straightness between hydrogels generated from the CTX or MED dECM (Appendix A).

### 3.2. MAPF Was Able to Deplete POIs from dECM Hydrogels

We next aimed to generate dECM hydrogels that can be specifically and reliably depleted of POIs; in this case, EMILIN1 and ZP3. EMILIN1 was selectively removed from bovine CTX and MED dECM hydrogels by MAPF by 97.25% and 95.67% in the cortex and medulla, respectively (Figure 1A). Using the same method, there was an 88.75% and 98.4% reduction in ZP3 in CTX and MED dECM hydrogels, respectively (Figure 1B). To ensure that our process of removing a POI was specific to that protein, we measured the abundance of COL4, a ubiquitous protein that has been shown to colocalize with EMILIN1 in human and murine skin, before and after depletion [37,38,39]. Following MAPF that targeted EMILIN1, a nominal 6.1% reduction of COL4 for CTX dECM and a 21.0% reduction of COL4 for MED dECM hydrogels was observed (Figure 1C). Similarly, selective depletion of ZP3 by MAPF resulted in a limited decrease in the abundance of COL4 (14.2% reduction in CTX dECM,18.0% reduction in MED dECM hydrogels depleted of ZP3) (Figure 1D). Therefore, compartment-specific dECM hydrogels can be depleted of POIs without significant variations in the abundance of associated matrisome proteins.

### 3.3. MAPF Depletion of EMILIN1 and ZP3 Altered Hydrogel Rigidity and Fiber Architecture

To understand the effects of depleting EMILIN1 or ZP3 from dECM hydrogels, we examined the physical and architectural features of these hydrogels. First, we performed nanoindentation (NI) on compartment-specific hydrogels before and after the depletion of EMILIN1 and ZP3. Interestingly, we found that depletion of EMILIN1 in both CTX and MED dECM hydrogels resulted in a significant increase in rigidity (Figure 2A). Prior to depletion, CTX dECM hydrogels had a Young’s modulus of 20.46 ± 10.38 Pa, which was increased to 39.20 ± 20.61 Pa post-depletion of EMILIN1. However, there was no significant change in the Young’s modulus when hydrogels were depleted of ZP3. Similarly, MED dECM hydrogels once depleted of EMILIN1 were significantly more rigid (MED dECM: 17.75 ± 3.98 Pa, MED dECM depleted of EMILIN1: 48.68 ± 22.77 Pa), and the rigidity was not significantly different when depleted of ZP3.

Next, we characterized fiber dynamics within CTX and MED dECM hydrogels following the depletion of EMILIN1 or ZP3. Width, length, straightness, and fiber angle-angle alignment were determined to be similar in EMILIN1-depleted and ZP3-depleted CTX dECM hydrogels compared to CTX dECM (Figure 2B,C). However, depleting either EMILIN1 or ZP3 from the MED dECM hydrogels resulted in altered fiber width and angle-angle alignment (Figure 2D). MED dECM hydrogels depleted of EMILIN1 or ZP3 had increased fiber width compared to the starting MED dECM hydrogels (MED dECM = 56.94, MED dECM depleted of EMILIN1 = 60.16, and MED dECM depleted of ZP3 = 61.79 pixel/μm). Additionally, depletion of EMILIN1 reduced the fiber angle alignment in MED dECM hydrogels, while depletion of ZP3 in these same hydrogels increased fiber–fiber angles (MED dECM = 92.01, MED dECM depleted of EMILIN1 = 87.35, and MED dECM depleted of ZP3 = 94.32 degrees). These data indicate that while there were no significant changes in the major collagen protein, COL4, within CTX and MED dECM hydrogel following the depletion of EMILIN1 or ZP3, there were some notable changes in the rigidity and architecture in the EMILIN1-depleted and ZP3-depleted hydrogels. 

### 3.4. Compartment-Specific dECM Hydrogels Depleted of EMILIN1 Supported Primary Murine Follicle Growth

To investigate the potential role of EMILIN1 in folliculogenesis, we chose to test compartment-specific hydrogels and then their depleted EMILIN1 counterparts in a follicle growth assay with murine follicles. Primary murine follicles (≈115 μm) were encapsulated in groups of 9–11 in one of five conditions: CTX dECM, CTX dECM depleted of EMILIN1, MED dECM, MED dECM depleted of EMILIN1, and control 0.5% alginate. Follicles were then cultured for 6 days, and survival and growth were assessed. The optically clear hydrogels enabled us to track the follicle health and diameter over time and growth was compared to 0.5% alginate encapsulated follicles (Figure 3A). Remarkably, follicles grown in dECM hydrogels depleted of EMILIN1 were significantly smaller in diameter on day 6 when compared to the CTX or MED dECM hydrogels. Follicles grown in CTX dECM hydrogels grew to an average diameter of 176.53 μm by day 6, while follicles grown in MED dECM hydrogels grew to an average diameter of 184.82 μm by day 6. These are both diameters that were significantly above the average alginate diameter of 162.05 μm, whereas the follicles grown in the two EMILIN1-depleted hydrogels averaged smaller diameters than those in alginate (158.05 μm and 146.79 μm) (Figure 3B).

All conditions remained above 80% survival through day 6 (Figure 3C). As follicles directly interacted with the biomaterial, in contrast to the alginate control, there were some observations of expansion and migration of follicular cells within dECM hydrogels. This expansion sometimes resulted in a rupturing phenotype where a cumulus–oocyte complex containing a morphologically normal oocyte was extruded from the follicle. These observations indicate that hydrogels derived from ovary-specific dECM can be modified to deplete POIs and used to monitor folliculogenesis in vitro.

## 4. Discussion

OTC then OTT is the only option for individuals with ovaries to cryopreserve their fertility and restore the ability to have a biological child if they are unable to freeze eggs prior to treatment. While there are significant benefits to this approach, there are limitations in long-term fertility preservation and endocrine restoration that a bioprosthetic ovary could alleviate [3,40]. Many patients that undergo OTC have metastatic disease and are unable to utilize OTT. Alternative technologies that improve in vitro follicle growth and maturation methods, such as 3D encapsulation or a scaffold design that supports isolated PMFs that are removed from cancerous tissue, are needed to advance the field and serve these patients. These technologies rely on a greater understanding of how the microenvironment can influence PMF quiescence, follicle growth, and follicle survival. The methods described aim to expand our toolbox to better understand critical proteins and properties within the follicle microenvironment.

Both physical and biochemical cues are generated by matrisome proteins [41,42]. However, the specific role of each matrisome protein is still not well understood, particularly in relation to follicle activation, growth, and survival. Moreover, the ovary uniquely undergoes cyclical remodeling during ovulation with major alterations in the extrafollicular microenvironment, including the matrisome [43]. As such, deepening our understanding of how the matrisome contributes to folliculogenesis is of interest to the reproductive biology community. Bovine ovaries are an ideal resource for ECM materials as they mimic human ovaries [35,44,45,46]. The cortex and medulla have unique physical and biochemical properties, and prior work has shown that ovarian tissue can be decellularized and used as a scaffold for in vitro culture of ovarian cells and follicles [34,47,48,49,50]. Two studies have utilized ovarian-derived dECM hydrogels to study in vitro follicle growth and survival [51,52]. These studies laid the groundwork for the use of dECM hydrogels in reproductive science. However, examination of follicle growth and survival in dECM hydrogels derived from cortex- or medulla-specific compartments where POIs are removed have not yet been reported. We built upon these studies to generate compartment-specific, optically clear hydrogels, allowing for precise measurements and imaging of follicles within the encapsulated environment over time. Moreover, this work describes a robust and reliable method for successful depletion of dECM hydrogels of POIs.

To test the ability of our technique to selectively deplete POIs from dECM hydrogels, we demonstrated targeted depletion of EMILIN1 and ZP3, two proteins functionally implicated in follicle growth and with differing protein abundance across the ovary. Protein expression of EMILIN1 in the porcine ovary peaks within the cortex where PMFs reside [33]. Lower EMILIN1 abundance also coincides with an increase in ZP3, which is expressed after follicle activation [28,53,54]. ZP3 is a protein exclusively found in the ovary surrounding oocytes. We selected this protein for depletion but not in vitro follicle growth studies because murine follicles will each contain an abundance of ZP3 proteins. EMILIN1, however, is largely extrafollicular and antagonizes TGFβ signaling, which has may work through SMADs to induce granulosa cell proliferation and PMF activation in mice [55]. Therefore, we hypothesized that depleting EMILIN1 would impact follicle growth in vitro. We quantified the abundance of EMILIN; ZP3; and a control protein, COL4. We found that there were no differences in the amount of EMILIN1 or ZP3 between the compartment-specific hydrogels. While the observed differences in bovine (mono-ovulatory species)-derived hydrogels described here are different from what was previously reported in porcine (poly-ovulatory species) ovaries, these observations may be species-specific and/or result from differences in mode of ovulation [56,57]. Work characterizing the matrisome of the bovine ovary is ongoing and will further enhance our understanding of these important species-specific differences. While we were not able to detect a difference in the abundance of POIs in our compartment-specific dECM hydrogels, there was a reduction, albeit not statistically significant, in fiber width between hydrogels.

After MAPF of POIs, we demonstrated a depletion of EMILIN1 by 97.25% and 95.6% and ZP3 by 88.75% and 98.4% within CTX and MED dECM hydrogels, respectively, while COL4 levels were maintained. Additionally, there were significant changes in width and angle of collagen fibers of MED dECM depleted of EMILIN1. Cellular behavior is known to vary depending on the alignment of matrisome fibers, so these observed alterations in fiber architecture suggest changes in the structure of the depleted hydrogel despite depletion in protein abundance not reaching statistical significance [58,59,60,61,62]. Further studies to determine downstream effects of these changes on cellular behavior will be needed to explore functionality of the observed phenotype. We hypothesize this observation of architectural differences is likely due to an increase in ECM compaction with the removal of ECM glycoproteins that often colocalizes with collagens, fibrins, and fibrillins. These results support that POIs can be selectively removed from compartment-specific ovarian hydrogels using a MAPF technique without loss of important associated proteins, providing a novel tool for the selective study of matrisome proteins in vitro.

Rigidity of the environment surrounding follicles has been shown to play an important role in activation and growth [42,63]. Many studies have independently shown that the rigidity of the follicular environment maintains quiescence and modulates follicle growth in vitro. For example, secondary murine follicles grown in alginate beads grow larger with decreasing alginate stiffness [42,63]. Additionally, murine ovaries cultured in vitro maintain oocyte dormancy under exogenous pressure, demonstrating that pressure plays an independent role in follicle activation [64]. There were no significant differences between the Young’s modulus of the two hydrogels CTX dECM and MED dECM. This was an unexpected finding as the ovarian cortex has previously been shown to be more rigid than the medulla [28]. However, the bovine cortical dECM has also been previously shown to be denser than the medulla. We hypothesize that using 5 mg/mL to fabricate both dECM hydrogels may have decreased the relative rigidity of the dECM by equalizing density of CTX dECM to MED dECM. Interestingly, EMILIN1 depletion from dECM hydrogels increased rigidity of the hydrogels, which likely results from changes in the compaction of the hydrogels or an increase in hydrogel swelling. These phenotypes will be the subject of future research and represent exciting applications of these hydrogels in studying functional significance of biochemical and biophysical cues in a tissue microenvironment.

Chiti et al. (2022) showed that ovarian-derived dECM hydrogels are not rigid enough to support follicle culture and require supplementation with alginate, an approach we replicated in this study [51]. Using an alginate-spiked dECM hydrogel, we successfully encapsulated and cultured primary murine follicles within different dECM hydrogel formulations. In these studies, follicles grew to significantly different diameters across conditions and grew largest in CTX and MED dECM hydrogels compared to alginate and dECM depleted of EMILIN1.

EMILIN1 is known to modulate cell behavior, growth factor availability, and ECM assembly [65,66]. The N-terminus of EMILIN1 inhibits pro-TGFβ proteolysis by binding to pro-TGFβ precursors and preventing proteolysis by furin convertases, thereby limiting TGFβ bioavailability [67]. EMILIN1 deficiency has also been associated with activation of non-canonical and canonical TGFβ signaling [68,69,70,71,72]. Independently, TGFβ signaling has been extensively shown to modulate folliculogenesis. Therefore, the observed differences in follicle growth dynamics may be due to alterations in downstream TGFβ signaling or integrin occupation. Interestingly, follicles grown in dECM hydrogels depleted of EMILIN1 were smaller in diameter than when compared to follicles grown in 0.5% alginate. Future studies elucidating the mechanism by which EMILIN1 contributes to follicle growth will be of interest. Using the materials and tools developed here will elucidate the mechanism by which EMILIN1 impacts follicle growth.

Although much research is needed before these materials can be translated for clinical use, the current study details a novel method for exploring matrisome-dependent contributions to follicle growth in vitro. While this proof-of-concept study shows the successful implementation of depleted hydrogels in studying follicle growth dynamics, it would be beneficial in the future to perform experiments with other POIs are depleted. Additionally, it will be important to expand our understanding of the effects of POIs on folliculogenesis by measuring estradiol production and determining oocyte quality in the future. With further understanding of the bovine and human ovarian matrisome and additional investigations beyond the in vitro proof of concept described here, we believe this tool will advance development of a future bioprosthetic ovary and expand fertility and hormone restoration options for more individuals. Additionally, this method is highly translatable to organ systems and tissue types.

## 5. Conclusions

Compartmentalization of the ovarian matrisome plays an important role in follicle activation and growth. Here, dECM hydrogels were fabricated from each compartment of the ovary and POIs were selectively depleted from dECM hydrogels such that the roles of these matrisome proteins can be studied in vitro. dECM hydrogels and selectively depleted hydrogels supplemented with alginate were optically clear and supported primary murine follicle growth and survival for at least 6 days, with differences in growth based on protein composition. This method of depleting dECM hydrogels is an important tool for studying matrisome proteins in the context of folliculogenesis for the future generation of a bioprosthetic ovary and will additionally expand our ability to study the physical and biochemical cues of matrisome proteins in a wide variety of tissue types.

## Figures and Tables

**Figure 1 bioengineering-11-00543-f001:**
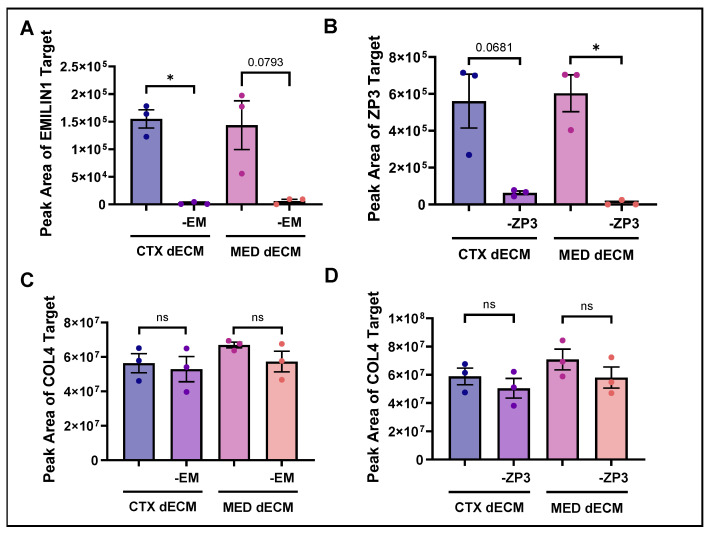
EMILIN1 and ZP3 were selectively removed from ovarian compartment-specific dECM hydrogels. (**A**) Amount of EMILIN1, (**B**) ZP3, or (**C**,**D**) COL4 detected by Western blot in cortex (CTX) or medulla (MED) dECM hydrogel following magnetic-assisted protein filtration of EMILIN1 (-EM) or ZP3 (-ZP3). Bars, mean +/− SEM, *, *p* < 0.05, ns, not significant.

**Figure 2 bioengineering-11-00543-f002:**
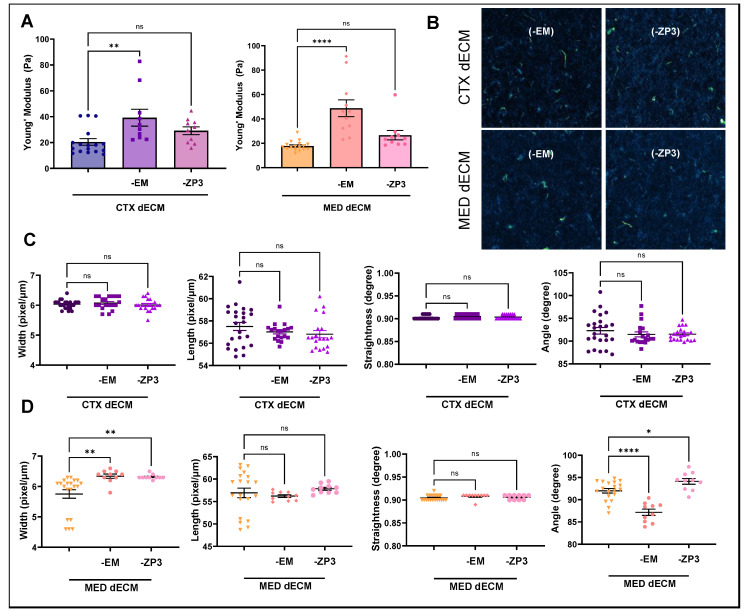
Characterization of depleted dECM hydrogels. (**A**) Rigidity of cortex (CTX) and medulla (MED) dECM hydrogels before and after depletion of EMILIN1 (-EM) or ZP3 (-ZP3) using nanoindentation. (**B**) Representative images of fiber organization in CTX and MED dECM hydrogels depleted of EMILIN1 (-EM) or ZP3 (-ZP3). (**C**) Analysis of fiber architecture in CTX and depleted CTX dECM hydrogels using CT-FIRE. Mean +/− SEM; ns, not significant. (**D**) Analysis of fiber architecture in MED and depleted MED dECM hydrogels using CT-FIRE. Mean +/− SEM *, **, **** *p* < 0.5, 0.005, 0.00005; ns, not significant.

**Figure 3 bioengineering-11-00543-f003:**
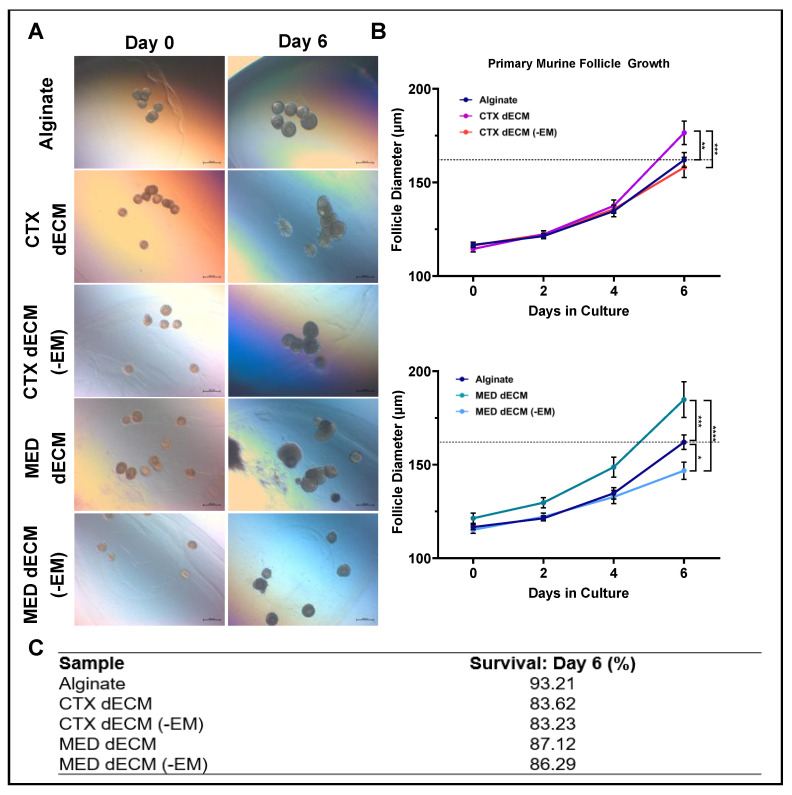
dECM hydrogels depleted of EMILIN1 support follicle growth and survival over a 6 day culture period. (**A**) Representative images of primary murine follicles encapsulated in alginate (control), CTX dECM, CTX dECM (-EM), MED dECM, or MED dECM (-EM) at day 0 and day 6. Scale = 200 μm. (**B**) Primary follicle growth curves representing 6 independent experiments with two beads of 10 +/− 1 follicle per experiment (*n* = 12). Beads containing <9 or >11 follicles were excluded. The dotted line denotes the average follicle diameter of the alginate group on day 6. Mean +/− SEM; *, **, ***, **** *p* value < 0.05, 0.005, 0.0003, and 0.0001. (**C**) Average percent of follicle survival on day 6 for each condition.

## Data Availability

The data presented in this study are available on request from the corresponding author.

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
