# Peer review of "Generation of Tailored Extracellular Matrix Hydrogels for the Study of In Vitro Folliculogenesis in Response to Matrisome-Dependent Biochemical Cues"

_bioengineering, 2024, doi:10.3390/bioengineering11060543_

Round 1

Reviewer 1 Report

Comments and Suggestions for Authors

This well-written article explores an interesting and key aspect of ovarian bioengineering and the OTC/OTT field. The methods developed here are also translatable to other areas, so I think this article is worth publishing. I want to emphasize the clarity and nice design of the figures—they make the reading much easier.

Still, I have minor comments for the authors before the article can be published:

1. I missed some information in the Introduction regarding the role/mechanism of EMILIN1 and ZP3 or at least the protein families to which they belong so the reader can have a bit of context.

2. Were all the ovaries collected in the same estrous cycle?

3. It could be a journal requirement, but I find it a bit unusual authors write Supplemental Figure X in the subheadings of Methods instead of along the text.

4. Methods, paragraph Ovarian Hydrogel Fabrication: I would remark that you are creating 2 different hydrogels (CTX and MED) - it is implied later, but it could help the reader if you include it here; I would also specify the final ECM concentration - once the digested ECM is neutralized and mixed with alginate.

5. Supplemental Fig. 1: misspelling error in "5 mg of cortex OR medulla".

6. Methods, paragraph "MAPF of ovarian hydrogels": At the beginning of the paragraph, I would remark that the POI depletion is done in the pre-gel form.

7. I have found some "results" in the "methods" and vice-versa. Eg.:

-Suppl. Fig. 3: These are results. Also, about this figure, I suggest reducing the Figure legend —what about instead of explaining 1 to 6 lanes 3 times, you replace -EM/-ZP for -POI and just write it once?

-Results, first paragraph: I would say this information belongs to methods or even introduction.

8. Figure 2B & Supp. Fig. 4B: what is the "green" marking?

9. I missed explaining why you depleted ZP3 but didn't perform folliculogenesis on those gels —you could add a few sentences about this to the discussion.

10. Authors mention in the discussion that they decided to create compartment-specific hydrogels. What are the implications/advantages vs. using the entire ovarian ECM all together?

11. I missed some self-criticism at the end of the discussion, how could this study be improved?

Author Response

The authors would like to thank the expert reviewers and editors for their time and efficiency in reviewing our manuscript for publication. Like the reviewers stated below, we hope that the tools represented in this manuscript are considered for research in other organ systems. We have addressed each point made below and hope that the editor will find the revised manuscript worthy of publication in Bioengineering.

Reviewer 1

This well-written article explores an interesting and key aspect of ovarian bioengineering and the OTC/OTT field. The methods developed here are also translatable to other areas, so I think this article is worth publishing. I want to emphasize the clarity and nice design of the figures—they make the reading much easier.

Still, I have minor comments for the authors before the article can be published:

  1. I missed some information in the Introduction regarding the role/mechanism of EMILIN1 and ZP3 or at least the protein families to which they belong so the reader can have a bit of context.

The authors have moved a portion of the results paragraph to the introduction (current version, lines 96 to 109). This paragraph explicitly states to the readers why EMILIN1 and ZP3 were selected for removal.

  1. Were all the ovaries collected in the same estrous cycle?

The authors are unsure whether the reviewer is asking about the bovine ovaries used for fabrication of the dECM hydrogels or the murine ovaries used for primary follicle culture. However, at the time dECM hydrogels were being fabricated, bovine ovaries were sourced by Assisted Reproductive Technologies (ART) which unfortunately could not provide cycle information on the cattle. Based on the gross morphology of the ovaries used in this study, we know that the cows were cycling due to the presence of large antral follicles and corpus luteum, but we are unable to ensure they are within the same estrous cycle. If the reviewer is referring to the murine ovaries used for the primary follicle culture, then the ovaries were collected in prepubertal mice of the same age which are not yet cycling. We have added a sentence to the material and methods section to clarify (lines 122 to 124).

  1. It could be a journal requirement, but I find it a bit unusual authors write Supplemental Figure X in the subheadings of Methods instead of along the text.

Thank you for letting the authors know. As the supplemental figure 1 and 2 are graphical representations of the method for generating dECM hydrogels (Sup Figure 1) and depleting hydrogels of proteins (Sup Figure 2) we felt it was more appropriate to add it to the subheading for the written method rather than in text.

  1. Methods, paragraph Ovarian Hydrogel Fabrication: I would remark that you are creating 2 different hydrogels (CTX and MED) - it is implied later, but it could help the reader if you include it here; I would also specify the final ECM concentration - once the digested ECM is neutralized and mixed with alginate.

Thank you for this insightful comment. We will update and specify that post neutralization, the final concentration of dECM is 4.16 mg/mL rather than the starting material concentration of 5 mg/mL. We have added additional detail to the methods section (lines 150-153).

  1. Supplemental Fig. 1: misspelling error in "5 mg of cortex ORmedulla".

The authors thank you for your diligent review. We have fixed this error.

  1. Methods, paragraph "MAPF of ovarian hydrogels": At the beginning of the paragraph, I would remark that the POI depletion is done in the pre-gel form.

The authors have added the specificity of MAPF depletion being performed pre-gelation in this paragraph.

  1. I have found some "results" in the "methods" and vice-versa. Eg.:

-Suppl. Fig. 3: These are results. Also, about this figure, I suggest reducing the Figure legend —what about instead of explaining 1 to 6 lanes 3 times, you replace -EM/-ZP for -POI and just write it once?

The authors appreciate the reviewer's suggestion for reducing the figure legend. We have updated the legend as suggested and have moved all supplemental figures to a “supplementary materials” section before author contributions.

-Results, first paragraph: I would say this information belongs to methods or even introduction.

Thank you for this recommendation. We agree that the majority of the first paragraph in the results would be better received if placed in the introduction. The authors have moved this paragraph to the introduction which also helps address reviewers' comments (1). By moving this paragraph, we introduce EMILIN1 and ZP3 to readers which provides more context as to why these proteins were selected. (See response to comment 1)

  1. Figure 2B & Supp. Fig. 4B: what is the "green" marking?

The green marking within the second harmonic generation microscopy is associated with a more intense signal. Using NIS Elements Viewer 5.21 64 bit the authors utilized the “Green Fire Blue” color setting in the LUTs window to highlight the intensity gradient of the fibers imaged. The authors have added information into the methods section for SHG (Lines 190-194).

  1. I missed explaining why you depleted ZP3 but didn't perform folliculogenesis on those gels —you could add a few sentences about this to the discussion.

The authors appreciate this question. Our goal of depleting multiple proteins (EMILIN1 and ZP3) from the dECM hydrogels was to show that this method can be used for a variety of matrisome proteins. ZP3 is a protein that is exclusively found in the ovary surrounding oocytes. As such we selected this protein for depletion but not in vitro follicle growth studies because the murine follicles will each contain an abundance of ZP3 proteins. EMILIN1 however, is largely extrafollicular (found surrounding follicles) and associated with antagonization of TGFB which has been shown to modulate folliculogenesis, so depletion of EMILIN1 in dECM hydrogels would directly impact in vitro follicle growth. Thank you for suggesting the inclusion of this detail.

  1. Authors mention in the discussion that they decided to create compartment-specific hydrogels. What are the implications/advantages vs. using the entire ovarian ECM all together?

The authors chose to create compartment-specific hydrogels as the Laronda Lab is interested in the differences in matrisome composition and abundance between the two compartments. The authors hypothesize that specific matrisome proteins presence and abundance may contribute to follicle quiescence and activation. We agree that it may be of interest to use the entire ovarian ECM in a “bulk” manner, and, based on the results described here, will likely use bulk ovarian ECM hydrogel material in future work.

  1. I missed some self-criticism at the end of the discussion, how could this study be improved?

The authors agree and have added to the discussion how this work could be improved in the future (lines 484 to 490). Thank you for your suggestion.  

Reviewer 2 Report

Comments and Suggestions for Authors

Comments from Reviewer

Generation of tailored extracellular matrix hydrogels for the study of in vitro folliculogenesis in response to matrisome-dependent biochemical cues

Hannah B. McDowell et al. 

Ovarian tissue cryopreservation (OTC) and ovarian tissue transplantation (OTT) are very promising technologies for fertility preservation. However, several issues remain to be explored. It is important to recapitulate the ovarian microenvironment on folliculogenesis.

As a novel tool for in vitro follicle growth, the authors have developed bovine-derived decellularized extracellular matrix (dECM) hydrogels that reflect the protein composition of the ovary. They generated the dECM hydrogels derived from bovine cortical tissue (CTX dECM) and medullary tissue (MED dECM). Furthermore, two specific proteins (EMILINI1 and ZP3) were selectively removed from the deCM hydrogel to determine the effects of these proteins on follicle growth in vitro. This study is unique and novel. The research strategies have been well organized. Proposed methods could be a promising model to understand the mechanisms of early-stage follicle growth and to improve the technologies related to OTT.

Major Comments:

(Comment 1) Figure 3B

The reviewer is concerned as to why follicle growth in the 2% alginate condition was better than that in the CTX dECM (-EM) or in the MED dECM (-EM) condition.  Even when depleting one protein (EMILIN1), the dECM (-EM) hydrogels still contain various ECM-derived bioactive factors; therefore the dECM (-EM) conditions seem more effective than the use of bioinert alginate on stimulating follicle growth. Please add a clear discussion on this point.

(Comment 2) Function of ovarian stromal cells

In this study, one-layered primary follicles encapsulated in the hydrogels were cultured without any contribution of ovarian stromal cells. Are the dECM hydrogels able to replace the ovarian stromal cells? How do the authors speculate the effects of the ovarian stromal cells on early-stage follicle growth? Please add discussion on this point.

Minor Comments:

(Comment 3) P6, L219 & P11, L316

Different follicle sizes (~110 μm, ~115 μm) were shown.

(Comment 4) P6, L238 ~ P7, 244 & Figure 3.

The authors measured follicle diameter and survival rates. They defined that ruptured follicles were classified as “viable” until the oocyte was more than 50% exposed. It is unclear what type of the follicles was used to measure the size. 

(Comment 5) P8, Figure 1. 

Show “B” and “D” on the graphs.

(Comment 6) P9, L292.

Modify as “there was no significant change”.

(Comment 7) 11, Figure 3B, top graph.

Colors of two bars (CTX dECM and CTX dECM (-EM)) are difficult to distinguish from each other. Please change the color(s).

Author Response

The authors would like to thank the expert reviewers and editors for their time and efficiency in reviewing our manuscript for publication. Like the reviewers stated below, we hope that the tools represented in this manuscript are considered for research in other organ systems. We have addressed each point made below and hope that the editor will find the revised manuscript worthy of publication in Bioengineering.

Reviewer 2

Comments from Reviewer

Generation of tailored extracellular matrix hydrogels for the study of in vitro folliculogenesis in response to matrisome-dependent biochemical cues

Hannah B. McDowell et al.

Ovarian tissue cryopreservation (OTC) and ovarian tissue transplantation (OTT) are very promising technologies for fertility preservation. However, several issues remain to be explored. It is important to recapitulate the ovarian microenvironment on folliculogenesis. As a novel tool for in vitro follicle growth, the authors have developed bovine-derived decellularized extracellular matrix (dECM) hydrogels that reflect the protein composition of the ovary. They generated the dECM hydrogels derived from bovine cortical tissue (CTX dECM) and medullary tissue (MED dECM). Furthermore, two specific proteins (EMILINI1 and ZP3) were selectively removed from the deCM hydrogel to determine the effects of these proteins on follicle growth in vitro. This study is unique and novel. The research strategies have been well organized. Proposed methods could be a promising model to understand the mechanisms of early-stage follicle growth and to improve the technologies related to OTT.

Major Comments:

(Comment 1) Figure 3B

The reviewer is concerned as to why follicle growth in the 2% alginate condition was better than that in the CTX dECM (-EM) or in the MED dECM (-EM) condition.  Even when depleting one protein (EMILIN1), the dECM (-EM) hydrogels still contain various ECM-derived bioactive factors; therefore the dECM (-EM) conditions seem more effective than the use of bioinert alginate on stimulating follicle growth. Please add a clear discussion on this point.

The authors would like to thank the reviewer for this important question. We hypothesized that the removal of EMILIN1 would result in follicles of smaller diameter; however, we were interested to see that the removal resulted in follicle diameters smaller than that of the (0.5%) alginate control. The are two possible mechanisms based on literature of EMILIN1 function in other tissues that may explain this phenomenon. EMILIN1 antagonizes TGFB signaling by inhibiting the pro-TGFB to TGFB conversion. Therefore, the removal of EMILIN1 in our models may ultimately increase TGFB within the culture. In addition to EMILIN1’s role in TGFB signaling, it has been shown to bind ITGA9 and ITGA4 exerting downstream signaling through integrin occupation. Integrin occupation results in increased cell proliferation. As such, the removal of EMILIN1 may result in a reduction in integrin occupation and potentially a reduction in granulosa cell proliferation. Therefore, the authors are not concerned that follicles grown in hydrogels depleted of EMILIN1 are smaller than those grown in 0.5% alginate. It may be of interest in the future to deplete proteins that we hypothesize would not impact follicle growth as a comparison.

Our main goal was not to produce hydrogels that are more effective than the control, but rather to study the effects of the removal of proteins on follicle growth. Finally, we aim to utilize this methodology to further investigate the mechanism by which EMILIN1 contributes to follicle growth in future work. However, elucidating the mechanism is currently out of the scope of this manuscript.

(Comment 2) Function of ovarian stromal cells

In this study, one-layered primary follicles encapsulated in the hydrogels were cultured without any contribution of ovarian stromal cells. Are the dECM hydrogels able to replace the ovarian stromal cells? How do the authors speculate the effects of the ovarian stromal cells on early-stage follicle growth? Please add discussion on this point.

This is an excellent question and an area of active investigation by the authors, though these ongoing experiments are outside the scope of this article. A recent published article comparing enzymatic isolation of follicles vs. mechanical isolation (which is the method used in this study), maintained stromal populations that may contribute to follicle growth seen in this study (Ref: Babayev, E. et al., Molecular Human Reproduction, https://doi.org/10.1093/molehr/gaac033). In future studies, we could test this more directly by comparing growth of follicles within these dECM hydrogels with or without enzymatic isolation to more directly evaluate the role of stromal cells in the context of this culture.

Minor Comments:

(Comment 3) P6, L219 & P11, L316

Different follicle sizes (~110 μm, ~115 μm) were shown.

The authors thank you for your thoroughness, we have updated the text to indicate the correct average starting diameter of 115 μm.

(Comment 4) P6, L238 ~ P7, 244 & Figure 3.

The authors measured follicle diameter and survival rates. They defined that ruptured follicles were classified as “viable” until the oocyte was more than 50% exposed. It is unclear what type of the follicles was used to measure the size.

Thank you for pointing out this critical step in determining follicle measurements. As seen in Figure 3A, specifically within the medullary day 6 image, there is a clear rupturing morphology of the follicle in the middle of the image when compared to others with a more defined basement membrane. Follicle diameters were assessed for follicles that maintained a basement membrane so that diameters were not inappropriately perceived as larger. The authors have added this point to the methods section to indicate which follicles were considered in the final measurements.

(Comment 5) P8, Figure 1.

Show “B” and “D” on the graphs.

The authors have fixed this error.

(Comment 6) P9, L292.

Modify as “there was no significant change”.

The authors have made this change.

(Comment 7) 11, Figure 3B, top graph.

Colors of two bars (CTX dECM and CTX dECM (-EM)) are difficult to distinguish from each other. Please change the color(s).

Thank you for identifying this error. We have changed the CTX dECM (-EM) to a red tone for ease of distinction.

Reviewer 3 Report

Comments and Suggestions for Authors

The authors have developed a dECM hydrogel method for the study of in vitro folliculogenesis. The topic is both current and interesting, and the manuscript is well-written, providing a valuable contribution to the design of an enhanced bioengineered scaffold for a bioprosthetic ovary. There are no major concerns with the manuscript. The reviewer believes that the manuscript is appropriate for publication in the Bioengineering journal with minor revisions. Proofreading is recommended.

The aim of the method you developed is to study of folliculogenesis, but there is insufficient information about the in vitro folliculogenesis. For example, only follicle growth and survival were estimated here.

Could you please explain what is meant by “grossly normal ovaries” as mentioned in your manuscript?

Line 123, page3, “H20” should be “H2O”.

Supplemental Figures should be placed in the supplemental materials section.

Letters B and C are missing in Figure 1.

Author Response

The authors would like to thank the expert reviewers and editors for their time and efficiency in reviewing our manuscript for publication. Like the reviewers stated below, we hope that the tools represented in this manuscript are considered for research in other organ systems. We have addressed each point made below and hope that the editor will find the revised manuscript worthy of publication in Bioengineering.

Reviewer 3

Comments and Suggestions for Authors

The authors have developed a dECM hydrogel method for the study of in vitro folliculogenesis. The topic is both current and interesting, and the manuscript is well-written, providing a valuable contribution to the design of an enhanced bioengineered scaffold for a bioprosthetic ovary. There are no major concerns with the manuscript. The reviewer believes that the manuscript is appropriate for publication in the Bioengineering journal with minor revisions. Proofreading is recommended.

The aim of the method you developed is to study of folliculogenesis, but there is insufficient information about in vitro folliculogenesis. For example, only follicle growth and survival were estimated here.

The authors used isolated murine follicles as an assay for this article because (1) there is a wealth of knowledge about isolated murine follicle growth rates and morphologies in encapsulated materials and (2) there are well-established methodologies for the isolation, encapsulation, and evaluation of murine follicle growth and survival. There is also precedent for this methodolfogy as the only other two articles that have used decellularized ECM for follicle encapsulation have utilized murine follicles for their proof-of-concept methodologies (Chiti et al. 2022; Frances-Herrero et al. 2023). Investigation of follicle growth and survival are standard output measurements for studying folliculogenesis in vitro. The authors agree it will be of interest to expand our understanding of the effects of depletion on folliculogenesis by measuring estradiol production and determining oocyte quality and we hope to follow up on this article with additional work. However, the goal of this article is to provide a novel tool for studying matrisome-dependent cues in vitro with the use of organ-specific dECM hydrogels depleted of proteins of interest and believe important knowledge was gained during this work.

Could you please explain what is meant by “grossly normal ovaries” as mentioned in your manuscript?

The authors would like to thank the reviewer for this question. The ovaries used to generate dECM for hydrogel fabrication were defined as “grossly normal” meaning they were of similar sizes, had antral follicles present, did not contain large corpus luteum, and displayed typical ovarian sub-anatomy. 

Line 123, page3, “H20” should be “H2O”.

Thank you, we have corrected this error.

Supplemental Figures should be placed in the supplemental materials section.

Thank you for confirming this. The authors have moved the supplemental figures to a “supplementary materials” section before authors contributions.

Letters B and C are missing in Figure 1.

Thank you, we have corrected this error.